# SVCCA: Singular Vector Canonical Correlation Analysis for Deep Learning Dynamics and Interpretability

**Maithra Raghu,**[1,2] **Justin Gilmer,**[1] **Jason Yosinski,**[3] **& Jascha Sohl-Dickstein**[1]
[1]Google Brain [2]Cornell University [3]Uber AI Labs
maithrar@gmail.com, gilmer@google.com, yosinski@uber.com, jaschasd@google.com

## Abstract

We propose a new technique, Singular Vector Canonical Correlation Analysis (SVCCA), a tool for quickly comparing two representations in a way that is both invariant to affine transform (allowing comparison between different layers and networks) and fast to compute (allowing more comparisons to be calculated than with previous methods). We deploy this tool to measure the intrinsic dimensionality of layers, showing in some cases needless over-parameterization; to probe learning dynamics throughout training, finding that networks converge to final representations from the bottom up; to show where class-specific information in networks is formed; and to suggest new training regimes that simultaneously save computation and overfit less.

## 1  Introduction

As the empirical success of deep neural networks ([7, 9, 18]) become an indisputable fact, the goal of better understanding these models escalates in importance. Central to this aim is a core issue of deciphering learned representations. Facets of this key question have been explored empirically, particularly for image models, in [1, 2, 10, 12, 13, 14, 15, 19, 20]. Most of these approaches are motivated by interpretability of learned representations. More recently, [11] studied the similarities of representations learned by multiple networks by finding permutations of neurons with maximal correlation.

In this work we introduce a new approach to the study of network representations, based on an analysis of each neuron's *activation vector* – the scalar outputs it emits on input datapoints. With this interpretation of neurons as vectors (and layers as subspaces, spanned by neurons), we introduce SVCCA, Singular Vector Canonical Correlation Analysis, an amalgamation of Singular Value Decomposition and Canonical Correlation Analysis (CCA) [5], as a powerful method for analyzing deep representations. Although CCA has not previously been used to compare deep representations, it has been used for related tasks such as computing the similarity between modeled and measured brain activity [16], and training multi-lingual word embeddings in language models [3].

The main contributions resulting from the introduction of SVCCA are the following:

1. We ask: is the dimensionality of a layer's learned representation the same as the number of neurons in the layer? *Answer: No.* We show that trained networks perform equally well with a number of directions just a fraction of the number of neurons with no additional training, provided they are carefully chosen with SVCCA (Section 2.1). We explore the consequences for model compression (Section 4.4).

2. We ask: what do deep representation learning dynamics look like? *Answer: Networks broadly converge bottom up.* Using SVCCA, we compare layers across time and find they

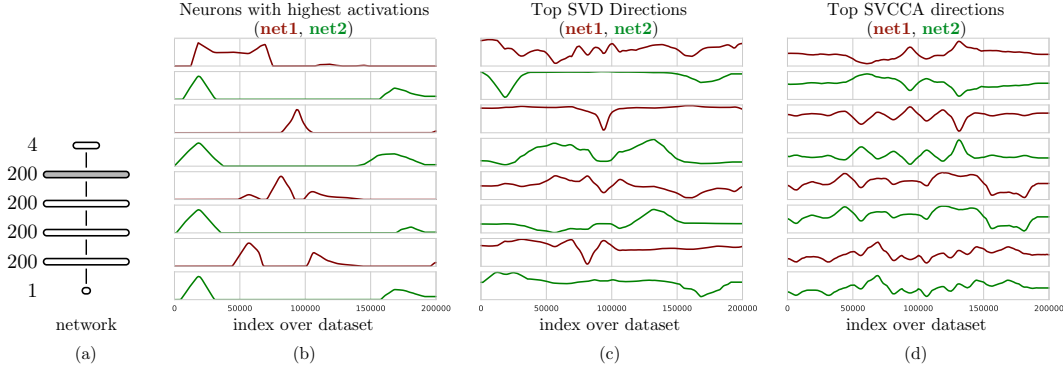

Figure 1: To demonstrate SVCCA, we consider a toy regression task (regression target as in Figure 3). **(a)** We train two networks with four fully connected hidden layers starting from different random initializations, and examine the representation learned by the penultimate (shaded) layer in each network. **(b)** The neurons with the highest activations in net 1 (maroon) and in net 2 (green). The x-axis indexes over the dataset: in our formulation, the *representation* of a neuron is simply its value over a dataset (Section 2). **(c)** The SVD directions — i.e. the directions of maximal variance — for each network. **(d)** The top SVCCA directions. We see that each pair of maroon/green lines (starting from the top) are almost visually identical (up to a sign). Thus, although looking at just neurons (b) seems to indicate that the networks learn very different representations, looking at the SVCCA subspace (d) shows that the information in the representations are (up to a sign) nearly identical.

solidify from the bottom up. This suggests a simple, computationally more efficient method of training networks, *Freeze Training*, where lower layers are sequentially frozen after a certain number of timesteps (Sections 4.1, 4.2).

3. We develop a method based on the discrete Fourier transform which greatly speeds up the application of SVCCA to convolutional neural networks (Section 3).

4. We also explore an interpretability question, of when an architecture becomes sensitive to different classes. We find that SVCCA captures the semantics of different classes, with similar classes having similar sensitivities, and vice versa. (Section 4.3).

**Experimental Details**   Most of our experiments are performed on CIFAR-10 (augmented with random translations). The main architectures we use are a convolutional network and a residual network[1]. To produce a few figures, we also use a toy regression task: training a four hidden layer fully connected network with 1D input and 4D output, to regress on four different simple functions.

## 2   Measuring Representations in Neural Networks

Our goal in this paper is to analyze and interpret the representations learned by neural networks. The critical question from which our investigation departs is: how should we define the representation of a neuron? Consider that a neuron at a particular layer in a network computes a real-valued function over the network's input domain. In other words, if we had a lookup table of all possible $\text{input} \rightarrow \text{output}$ mappings for a neuron, it would be a complete portrayal of that neuron's functional form.

However, such infinite tables are not only practically infeasible, but are also problematic to process into a set of conclusions. Our primary interest is not in the neuron's response to random data, but rather in how it represents features of a specific dataset (e.g. natural images). Therefore, in this study we take *a neuron's representation to be its set of responses over a finite set of inputs* — those drawn from some training or validation set.

More concretely, for a given dataset $X = \{x_1, \cdots x_m\}$ and a neuron $i$ on layer $l$, $\boldsymbol{z}_i^l$, we *define* $\boldsymbol{z}_i^l$ to be the *vector* of outputs on $X$, i.e.

$$\boldsymbol{z}_i^l = (\boldsymbol{z}_i^l(x_1), \cdots, \boldsymbol{z}_i^l(x_m))$$

Note that this is a different vector from the often-considered vector of the "representation at a layer of a single input." Here $z_i^l$ is a *single* neuron's response over the entire dataset, not an entire layer's response for a single input. In this view, a neuron's representation can be thought of as a single vector in a high-dimensional space. Broadening our view from a single neuron to the collection of neurons in a layer, the layer can be thought of as the set of neuron vectors contained within that layer. This set of vectors will span some subspace. To summarize:

> *Considered over a dataset $X$ with $m$ examples, a neuron is a vector in $\mathbb{R}^m$.*
> *A layer is the subspace of $\mathbb{R}^m$ spanned by its neurons' vectors.*

Within this formalism, we introduce *Singular Vector Canonical Correlation Analysis* (SVCCA) as a method for analysing representations. SVCCA proceeds as follows:

- **Input:** SVCCA takes as input two (not necessarily different) sets of neurons (typically layers of a network) $l_1 = \{z_1^{l_1}, ..., z_{m_1}^{l_1}\}$ and $l_2 = \{z_1^{l_2}, ..., z_{m_2}^{l_2}\}$
- **Step 1** First SVCCA performs a singular value decomposition of each subspace to get subspaces $l_1' \subset l_1, l_2' \subset l_2$ which comprise of the most important directions of the original subspaces $l_1, l_2$. In general we take enough directions to explain 99% of variance in the subspace. This is especially important in neural network representations, where as we will show many low variance directions (neurons) are primarily noise.
- **Step 2** Second, compute the Canonical Correlation similarity ([5]) of $l_1', l_2'$: linearly transform $l_1', l_2'$ to be as aligned as possible and compute correlation coefficients. In particular, given the output of step 1, $l_1' = \{z'_1^{l_1}, ..., z'_{m_1'}^{l_1}\}, l_2' = \{z'_1^{l_2}, ..., z'_{m_2'}^{l_2}\}$, CCA linearly transforms these subspaces $\tilde{l}_1 = W_X l_1', \tilde{l}_2 = W_Y l_2'$ such as to maximize the correlations $corrs = \{\rho_1, \ldots \rho_{\min(m_1', m_2')}\}$ between the transformed subspaces.
- **Output:** With these steps, SVCCA outputs pairs of aligned directions, $(\tilde{z}_i^{l_1}, \tilde{z}_i^{l_2})$ and how well they correlate, $\rho_i$. Step 1 also produces intermediate output in the form of the top singular values and directions.

For a more detailed description of each step, see the Appendix. SVCCA can be used to analyse any two sets of neurons. In our experiments, we utilize this flexibility to compare representations across different random initializations, architectures, timesteps during training, and specific classes and layers.

Figure 1 shows a simple, intuitive demonstration of SVCCA. We train a small network on a toy regression task and show each step of SVCCA, along with the resulting very similar representations. SVCCA is able to find hidden similarities in the representations.

## 2.1 Distributed Representations

An important property of SVCCA is that it is truly a *subspace* method: both SVD and CCA work with $\text{span}(z_1, \ldots, z_m)$ instead of being axis aligned to the $z_i$ directions. SVD finds singular vectors $z_i' = \sum_{j=1}^m s_{ij} z_j$, and the subsequent CCA finds a linear transform $W$, giving orthogonal canonically correlated directions $\{\tilde{z}_1, \ldots, \tilde{z}_m\} = \{\sum_{j=1}^m w_{1j} z_j', \ldots, \sum_{j=1}^m w_{mj} z_j'\}$. In other words, SVCCA has no preference for representations that are neuron (axes) aligned.

If representations are distributed across many dimensions, then this is a desirable property of a representation analysis method. Previous studies have reported that representations may be more complex than either fully distributed or axis-aligned [17, 21, 11] but this question remains open.

We use SVCCA as a tool to probe the nature of representations via two experiments:

(a) We find that the subspace directions found by SVCCA are disproportionately important to the representation learned by a layer, relative to neuron-aligned directions.

(b) We show that at least some of these directions are distributed across many neurons.

Experiments for (a), (b) are shown in Figure 2 as (a), (b) respectively. For both experiments, we first acquire two different representations, $l_1, l_2$, for a layer $l$ by training two different random initializations of a convolutional network on CIFAR-10. We then apply SVCCA to $l_1$ and $l_2$ to get directions

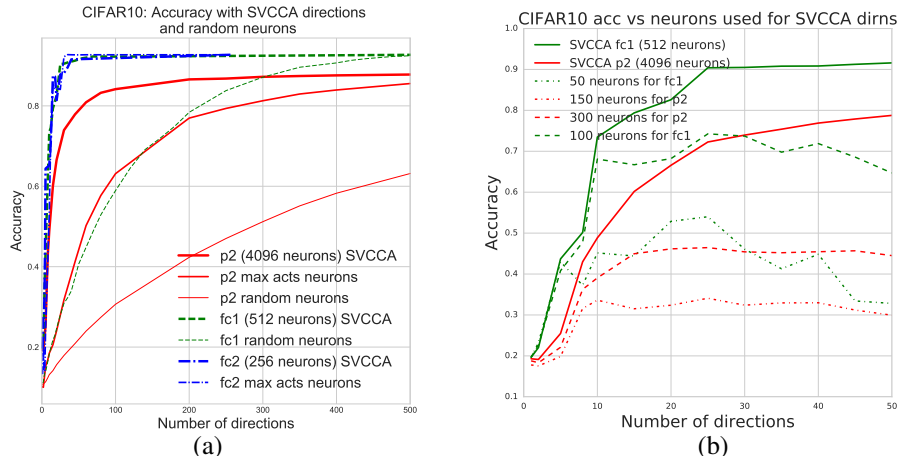

(a)  (b)

Figure 2: Demonstration of *(a)* disproportionate importance of SVCCA directions, and *(b)* distributed nature of some of these directions. For both panes, we first find the top $k$ SVCCA directions by training two conv nets on CIFAR-10 and comparing corresponding layers. *(a)* We project the output of the top three layers, pool1, fc1, fc2, onto this top-$k$ subspace. We see accuracy rises rapidly with increasing $k$, with even $k \ll$ num neurons giving reasonable performance, with *no* retraining. Baselines of random $k$ neuron subspaces and max activation neurons require larger $k$ to perform as well. *(b)*: after projecting onto top $k$ subspace (like left), dotted lines then project again onto $m$ neurons, chosen to correspond highly to the top $k$-SVCCA subspace. Many more neurons are needed than $k$ for better performance, suggesting distributedness of SVCCA directions.

$\{\tilde{z}_1^{l_1}, ..., \tilde{z}_m^{l_1}\}$ and $\{\tilde{z}_1^{l_2}, ..., \tilde{z}_m^{l_2}\}$, ordered according to importance by SVCCA, with each $\tilde{z}_j^{l_i}$ being a linear combination of the original neurons, i.e. $\tilde{z}_j^{l_i} = \sum_{r=1}^m \alpha_{jr}^{(l_i)} z_r^{l_i}$.

For different values of $k < m$, we can then restrict layer $l_i$'s output to lie in the subspace of $\mathrm{span}(\tilde{z}_1^{l_i}, \ldots, \tilde{z}_k^{l_i})$, the most useful $k$-dimensional subspace as found by SVCCA, done by projecting each neuron into this $k$ dimensional space.

We find — somewhat surprisingly — that very few SVCCA directions are required for the network to perform the task well. As shown in Figure 2(a), for a network trained on CIFAR-10, the first 25 dimensions provide nearly the same accuracy as using all 512 dimensions of a fully connected layer with 512 neurons. The accuracy curve rises rapidly with the first few SVCCA directions, and plateaus quickly afterwards, for $k \ll m$. This suggests that the useful information contained in $m$ neurons is well summarized by the subspace formed by the top $k$ SVCCA directions. Two baselines for comparison are picking random and maximum activation neuron aligned subspaces and projecting outputs onto these. Both of these baselines require far more directions (in this case: neurons) before matching the accuracy achieved by the SVCCA directions. These results also suggest approaches to model compression, which are explored in more detail in Section 4.4.

Figure 2(b) next demonstrates that these useful SVCCA directions are at least somewhat distributed over neurons rather than axis-aligned. First, the top $k$ SVCCA directions are picked and the representation is projected onto this subspace. Next, the representation is further projected onto $m$ neurons, where the $m$ are chosen as those most important to the SVCCA directions . The resulting accuracy is plotted for different choices of $k$ (given by x-axis) and different choices of $m$ (different lines). That, for example, keeping even 100 fc1 neurons (dashed green line) cannot maintain the accuracy of the first 20 SVCCA directions (solid green line at x-axis 20) suggests that those 20 SVCCA directions are distributed across 5 or more neurons each, on average. Figure 3 shows a further demonstration of the effect on the output of projecting onto top SVCCA directions, here for the toy regression case.

**Why the two step SV + CCA method is needed.** Both SVD and CCA have important properties for analysing network representations and SVCCA consequently benefits greatly from being a two step method. CCA is **invariant** to affine transformations, enabling comparisons without natural alignment (e.g. different architectures, Section 4.4). See Appendix B for proofs and a demonstrative figure. While CCA is a powerful method, it also suffers from certain shortcomings, particularly in determining how many directions were important to the original space $X$, which is the strength of

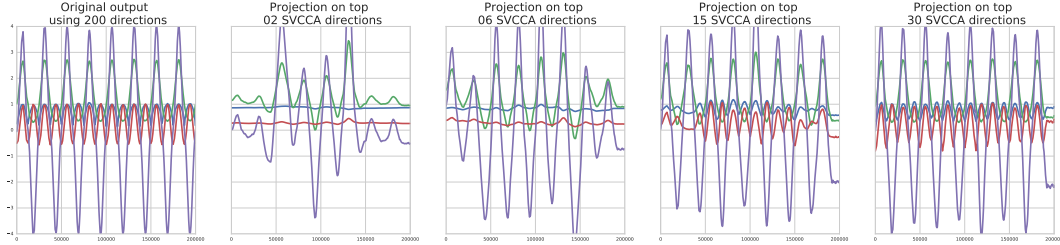

Figure 3: The effect on the output of a latent representation being projected onto top SVCCA directions in the toy regression task. Representations of the penultimate layer are projected onto $2, 6, 15, 30$ top SVCCA directions (from second pane). By 30, the output looks very similar to the full 200 neuron output (left).

SVD. See Appendix for an example where naive CCA performs badly. Both the SVD and CCA steps are critical to the analysis of learning dynamics in Section 4.1.

# 3 Scaling SVCCA for Convolutional Layers

Applying SVCCA to convolutional layers can be done in two natural ways:

(1) *Same layer comparisons:* If $X, Y$ are the same layer (at different timesteps or across random initializations) receiving the same input we can concatenate along the pixel (height $h$, width $w$) coordinates to form a vector: a conv layer $h \times w \times c$ maps to $c$ vectors, each of dimension $hwd$, where $d$ is the number of datapoints. This is a natural choice because neurons at different pixel coordinates see *different* image data patches to each other. When $X, Y$ are two versions of the same layer, these $c$ different views correspond perfectly.

(2) *Different layer comparisons:* When $X, Y$ are not the same layer, the image patches seen by different neurons have no natural correspondence. But we can flatten an $h \times w \times c$ conv into $hwc$ neurons, each of dimension $d$. This approach is valid for convs in different networks or at different depths.

## 3.1 Scaling SVCCA with Discrete Fourier Transforms

Applying SVCCA to convolutions introduces a computational challenge: the number of neurons ($h \times w \times c$) in convolutional layers, especially early ones, is very large, making SVCCA prohibitively expensive due to the large matrices involved. Luckily the problem of approximate dimensionality reduction of large matrices is well studied, and efficient algorithms exist, e.g. [4].

For convolutional layers however, we can avoid dimensionality reduction and perform *exact* SVCCA, even for large networks. This is achieved by preprocessing each channel with a Discrete Fourier Transform (which preserves CCA due to invariances, see Appendix), causing all (covariance) matrices to be block-diagonal. This allows all matrix operations to be performed block by block, and only over the diagonal blocks, vastly reducing computation. We show:

**Theorem 1.** *Suppose we have a translation invariant (image) dataset $X$ and convolutional layers $l_1$, $l_2$. Letting $DFT(l_i)$ denote the discrete fourier transform applied to each channel of $l_i$, the covariance $cov(DFT(l_1), DFT(l_2))$ is block diagonal, with blocks of size $c \times c$.*

We make only two assumptions: 1) all layers below $l_1$, $l_2$ are either conv or pooling layers with circular boundary conditions (translation equivariance) 2) The dataset $X$ has all translations of the images $X_i$. This is necessary in the proof for certain symmetries in neuron activations, but these symmetries typically exist in natural images even without translation invariance, as shown in Figure App.2 in the Appendix. Below are key statements, with proofs in Appendix.

**Definition 1.** *Say a single channel image dataset $X$ of images is translation invariant if for any (wlog $n \times n$) image $X_i \in X$, with pixel values $\{z_{11}, ...z_{nn}\}$, $X_i^{(a,b)} = \{z_{\sigma_a(1)\sigma_b(1)}, ...z_{\sigma_a(n)\sigma_b(n)}\}$ is also in $X$, for all $0 \le a, b \le n - 1$, where $\sigma_a(i) = a + i \mod n$ (and similarly for b).*

*For a multiple channel image $X_i$, an $(a, b)$ translation is an $(a, b)$ height/width shift on every channel separately. $X$ is then translation invariant as above.*

To prove Theorem 1, we first show another theorem:

**Theorem 2.** *Given a translation invariant dataset $X$, and a convolutional layer $l$ with channels $\{c_1, \ldots c_k\}$ applied to $X$*

*(a) the DFT of $c_i$, $FcF^T$ has diagonal covariance matrix (with itself).*

*(b) the DFT of $c_i, c_j$, $Fc_iF^T$, $Fc_jF^T$ have diagonal covariance with each other.*

Finally, both of these theorems rely on properties of *circulant matrices* and their DFTs:

**Lemma 1.** *The covariance matrix of $c_i$ applied to translation invariant $X$ is circulant and block circulant.*

**Lemma 2.** *The DFT of a circulant matrix is diagonal.*

# 4 Applications of SVCCA

## 4.1 Learning Dynamics with SVCCA

We can use SVCCA as a window into learning dynamics by comparing the representation at a layer at different points during training to its final representation. Furthermore, as the SVCCA computations are relatively cheap to compute compared to methods that require training an auxiliary network for each comparison [1, 10, 11], we can compare all layers during training at all timesteps to all layers at the final time step, producing a rich view into the learning process.

The outputs of SVCCA are the aligned directions $(\tilde{x}_i, \tilde{y}_i)$, how well they align, $\rho_i$, as well as intermediate output from the first step, of singular values and directions, $\lambda_X^{(i)}, x'^{(i)}, \lambda_Y^{(j)}, y'^{(j)}$. We condense these outputs into a single value, the *SVCCA similarity* $\bar{\rho}$, that encapsulates how well the representations of two layers are aligned with each other,

$$\bar{\rho} = \frac{1}{\min{(m_1, m_2)}} \sum_i \rho_i, \tag{1}$$

where $\min{(m_1, m_2)}$ is the size of the smaller of the two layers being compared. The SVCCA similarity $\bar{\rho}$ is the average correlation across aligned directions, and is a direct multidimensional analogue of Pearson correlation.

The SVCCA similarity for all pairs of layers, and all time steps, is shown in Figure 4 for a convnet and a resnet architecture trained on CIFAR10.

## 4.2 Freeze Training

Observing in Figure 4 that networks broadly converge from the bottom up, we propose a training method where we successively *freeze* lower layers during training, only updating higher and higher layers, saving *all* computation needed for deriving gradients and updating in lower layers.

We apply this method to convolutional and residual networks trained on CIFAR-10, Figure 5, using a linear freezing regime: in the convolutional network, each layer is frozen at a fraction (layer number/total layers) of total training time, while for resnets, each residual block is frozen at a fraction (block number/total blocks). The vertical grey dotted lines show which steps have another set of layers frozen. Aside from saving computation, Freeze Training appears to actively *help* generalization accuracy, like early stopping but with different layers requiring different stopping points.

## 4.3 Interpreting Representations: when are classes learned?

We also can use SVCCA to compare how correlated representations in each layer are with the logits of each class in order to measure how knowledge about the target evolves throughout the network. In Figure 6 we apply the DFT CCA technique on the Imagenet Resnet [6]. We take five different classes and for different layers in the network, compute the DFT CCA similarity between the logit of that class and the network layer. The results successfully reflect semantic aspects of the classes: the firetruck class sensitivity line is clearly distinct from the two pairs of dog breeds, and network develops greater sensitivity to firetruck earlier on. The two pairs of dog breeds, purposefully chosen so that each pair is similar to the other in appearance, have cca similarity lines that are very close to each other through the network, indicating these classes are similar to each other.

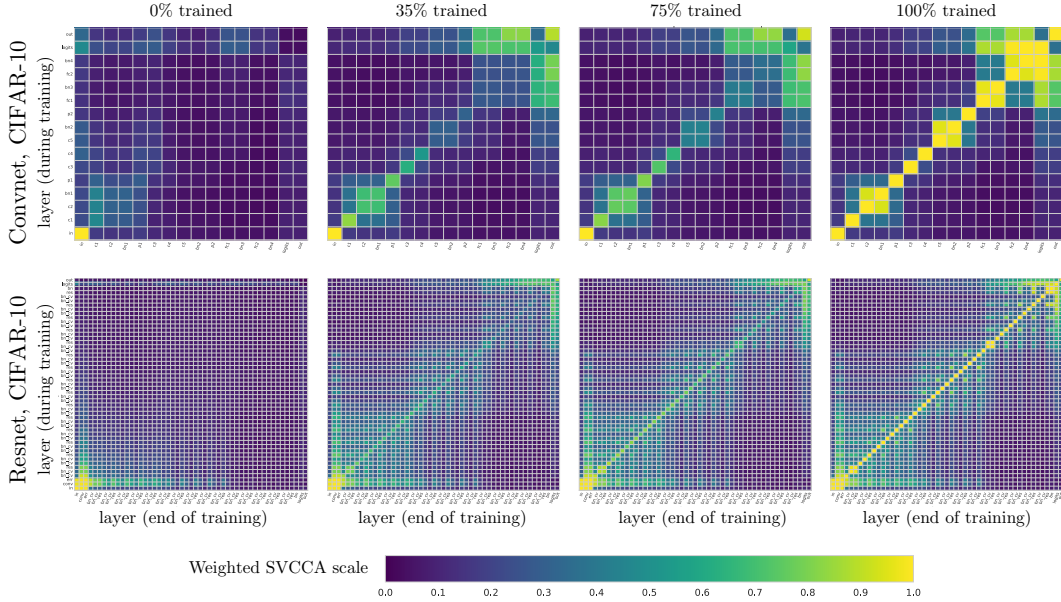

Figure 4: Learning dynamics plots for conv (top) and res (bottom) nets trained on CIFAR-10. Each pane is a matrix of size layers × layers, with each entry showing the SVCCA similarity $\bar{\rho}$ between the two layers. Note that learning broadly happens 'bottom up' – layers closer to the input seem to solidify into their final representations with the exception of the very top layers. Per layer plots are included in the Appendix. Other patterns are also visible – batch norm layers maintain nearly perfect similarity to the layer preceding them due to scaling invariance (with a slight reduction since batch norm changes the SVD directions which capture 99% of the variance). In the resnet plot, we see a stripe like pattern due to skip connections inducing high similarities to previous layers.

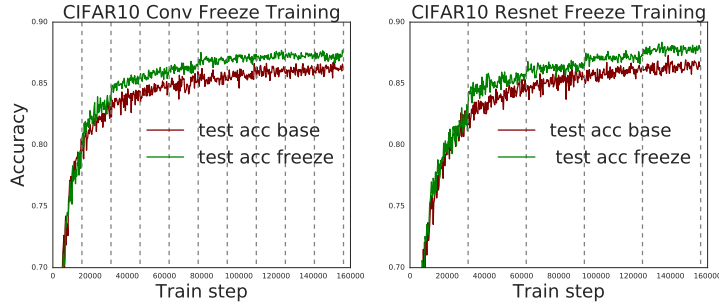

Figure 5: Freeze Training reduces training cost and improves generalization. We apply Freeze Training to a convolutional network on CIFAR-10 and a residual network on CIFAR-10. As shown by the grey dotted lines (which indicate the timestep at which another layer is frozen), both networks have a 'linear' freezing regime: for the convolutional network, we freeze individual layers at evenly spaced timesteps throughout training. For the residual network, we freeze entire residual blocks at each freeze step. The curves were averaged over ten runs.

## 4.4 Other Applications: Cross Model Comparison and compression

SVCCA similarity can also be used to compare the similarity of representations across different random initializations, and even different architectures. We compare convolutional networks on CIFAR-10 across random initializations (Appendix) and also a convolutional network to a residual network in Figure 7, using the DFT method described in 3.

In Figure 3, we saw that projecting onto the subspace of the top few SVCCA directions resulted in comparable accuracy. This observations motivates an approach to model compression. In particular, letting the output vector of layer $l$ be $\boldsymbol{x}^{(l)} \in \mathbb{R}^{n \times 1}$, and the weights $W^{(l)}$, we replace the usual $W^{(l)}\boldsymbol{x}^{(l)}$ with $(W^{(l)}P_x^T)(P_x\boldsymbol{x}^{(l)})$ where $P_x$ is a $k \times n$ projection matrix, projecting $\boldsymbol{x}$ onto the top SVCCA directions. This bottleneck reduces both parameter count and inference computational cost

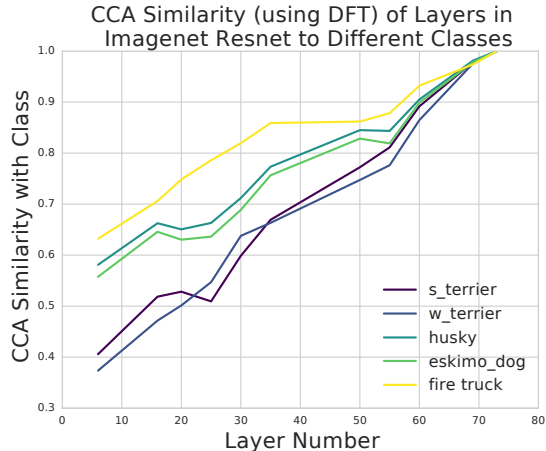

Figure 6: We plot the CCA similarity using the Discrete Fourier Transform between the logits of five classes and layers in the Imagenet Resnet. The classes are firetruck and two pairs of dog breeds (terriers and husky like dogs: husky and eskimo dog) that are chosen to be similar to each other. These semantic properties are captured in CCA similarity, where we see that the line corresponding to firetruck is clearly distinct from the two pairs of dog breeds, and the two lines in each pair are both very close to each other, reflecting the fact that each pair consists of visually similar looking images. Firetruck also appears to be *easier* for the network to learn, with greater sensitivity displayed much sooner.

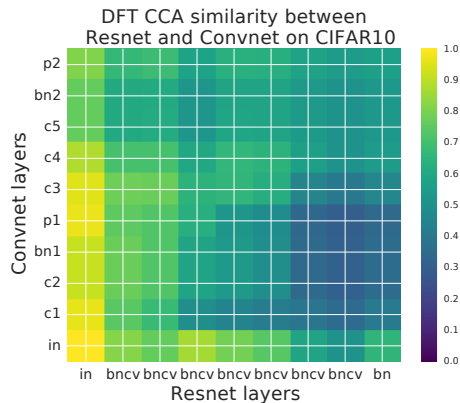

Figure 7: We plot the CCA similarity using the Discrete Fourier Transform between convolutional layers of a Resnet and Convnet trained on CIFAR-10. We find that the lower layrs of both models are noticeably similar to each other, and get progressively less similar as we compare higher layers. Note that the highest layers of the resnet are least similar to the lower layers of the convnet.

for the layer by a factor $\sim \frac{k}{n}$. In Figure App.5 in the Appendix, we show that we can *consecutively* compress top layers with SVCCA by a significant amount (in one case reducing each layer to $0.35$ original size) and hardly affect performance.

## 5 Conclusion

In this paper we present SVCCA, a general method which allows for comparison of the learned distributed representations between different neural network layers and architectures. Using SVCCA we obtain novel insights into the learning dynamics and learned representations of common neural network architectures. These insights motivated a new Freeze Training technique which can reduce the number of flops required to train networks and potentially even increase generalization performance. We observe that CCA similarity can be a helpful tool for interpretability, with sensitivity to different classes reflecting their semantic properties. This technique also motivates a new algorithm for model compression. Finally, the "lower layers learn first" behavior was also observed for recurrent neural networks as shown in Figure App.6 in the Appendix.

## Footnotes

[1]Convnet layers: `conv-conv-bn-pool-conv-conv-conv-bn-pool-fc-bn-fc-bn-out`. Resnet layers: `conv-(x10 c/bn/r block)-(x10 c/bn/r block)-(x10 c/bn/r block)-bn-fc-out`.

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
