[Supplementary Material]

# Appendix

## A    Mathematical details of CCA and SVCCA

**Canonical Correlation of** $X, Y$    Finding maximal correlations between $X, Y$ can be expressed as finding $a, b$ to maximise:

$$\frac{a^T \Sigma_{XY} b}{\sqrt{a^T \Sigma_{XX} a} \sqrt{b^T \Sigma_{YY} b}}$$

where $\Sigma_{XX}, \Sigma_{XY}, \Sigma_{YX}, \Sigma_{YY}$ are the covariance and cross-covariance terms. By performing the change of basis $\tilde{\boldsymbol{x}}_1 = \Sigma_{xx}^{1/2} a$ and $\tilde{\boldsymbol{y}}_1 = \Sigma_{YY}^{1/2} b$ and using Cauchy-Schwarz we recover an eigenvalue problem:

$$\tilde{\boldsymbol{x}}_1 = \text{argmax} \left[ \frac{x^T \Sigma_{XX}^{-1/2} \Sigma_{XY} \Sigma_{YY}^{-1} \Sigma_{YX} \Sigma_{XX}^{-1/2} x}{||x||} \right] \qquad (*)$$

**SVCCA**    Given two subspaces $X = \{\boldsymbol{x}_1, ..., \boldsymbol{x}_{m_1}\}, Y = \{\boldsymbol{y}_1, ..., \boldsymbol{y}_{m_2}\}$, SVCCA first performs a singular value decomposition on $X, Y$. This results in singular vectors $\{\boldsymbol{x'}_1, ..., \boldsymbol{x'}_{m_1}\}$ with associated singular values $\{\lambda_1, ..., \lambda_{m_1}\}$ (for $X$, and similarly for $Y$). Of these $m_1$ singular vectors, we keep the top $m_1'$ where $m_1'$ is the smallest value that $\sum_{i=1}^{m_1'} |\lambda_i| (\geq 0.99 \sum_{i=1}^{m_1} |\lambda_i|)$. That is, 99% of the variation of $X$ is explainable by the top $m_1'$ vectors. This helps remove directions/neurons that are constant zero, or noise with small magnitude.

Then, we apply Canonical Correlation Analysis (CCA) to the sets $\{\boldsymbol{x'}_1, ..., \boldsymbol{x'}_{m_1'}\}, \{\boldsymbol{y'}_1, ..., \boldsymbol{y'}_{m_2'}\}$ of top singular vectors.

CCA is a well established statistical method for understanding the similarity of two different sets of random variables – given our two sets of vectors $\{\boldsymbol{x'}_1, ..., \boldsymbol{x'}_{m_1'}\}, \{\boldsymbol{y'}_1, ..., \boldsymbol{y'}_{m_2'}\}$, we wish to find linear transformations, $W_X, W_Y$ that maximally correlate the subspaces. This can be reduced to an eigenvalue problem. Solving this results in linearly transformed subspaces $\tilde{X}, \tilde{Y}$ with directions $\tilde{\boldsymbol{x}}_i, \tilde{\boldsymbol{y}}_i$ that are maximally correlated with each other, and orthogonal to $\tilde{\boldsymbol{x}}_j, \tilde{\boldsymbol{y}}_j, j < i$. We let $\rho_i = corr(\tilde{\boldsymbol{x}}_i, \tilde{\boldsymbol{y}}_i)$. In summary, we have:

**SVCCA Summary**

1. Input: $X, Y$
2. Perform: SVD(X), SVD(Y). Output: $X' = UX, Y' = VY$
3. Perform CCA($X'$, $Y'$). Output: $\tilde{X} = W_X X', \tilde{Y} = W_Y Y'$ and $corrs = \{\rho_1, \cdots \rho_{\min(m_1, m_2)}\}$

## B    Additional Proofs and Figures from Section 2.1

Proof of Orthonormal and Scaling Invariance of CCA:

We can see this using equation (*) as follows: suppose $U, V$ are orthonormal transforms applied to the sets $X, Y$. Then it follows that $\Sigma_{XX}^a$ becomes $U \Sigma_{XX}^a U^T$, for $a = \{1, -1, 1/2, -1/2\}$, and similarly for $Y$ and $V$. Also note $\Sigma_{XY}$ becomes $U \Sigma_{XY} V^T$. Equation (*) then becomes

$$\tilde{x}_1 = \text{argmax} \left[ \frac{x^T U \Sigma_{XX}^{-1/2} \Sigma_{XY} \Sigma_{YY}^{-1} \Sigma_{YX} \Sigma_{XX}^{-1/2} U^T x}{||x||} \right]$$

So if $\tilde{u}$ is a solution to equation (*), then $U\tilde{u}$ is a solution to the equation above, which results in the same correlation coefficients.

### B.0.1    The importance of SVD: how many directions matter?

While CCA is excellent at identifying useful learned directions that correlate, independent of certain common transforms, it doesn't capture the full picture entirely. Consider the following setting:

Figure App.1: This figure shows the ability of CCA to deal with orthogonal and scaling transforms. In the first pane, the maroon plot shows one of the highest activation neurons in the penultimate layer of a network trained on CIFAR-10, with the x-axis being (ordered) image ids and the y-axis being activation on that image. The green plots show two resulting distorted directions after this and two of the other top activation neurons are permuted, rotated and scaled. Pane two shows the result of applying CCA to the distorted directions and the original signal, which succeeds in recovering the original signal.

suppose we have subspaces $A, B, C$, with $A$ being 50 dimensions, $B$ being 200 dimensions, 50 of which are perfectly aligned with $A$ and the other 150 being noise, and C being 200 dimensions, 50 of which are aligned with $A$ (and $B$) and the other 150 being useful, but different directions.

Then looking at the canonical correlation coefficients of $(A, B)$ and $(A, C)$ will give the *same* result, both being 1 for 50 values and 0 for everything else. But these are two very different cases – the subspace $B$ is indeed well represented by the 50 directions that are aligned with $A$. But the subspace $C$ has 150 more useful directions.

This distinction becomes particularly important when aggregating canonical correlation coefficients as a measure of similarity, as used in analysing network learning dynamics. However, by first applying SVD to determine the number of directions needed to explain $99\%$ of the observed variance, we can distinguish between pathological cases like the one above.

## C   Proof of Theorem 1

Here we provide the proofs for Lemma 1, Lemma 2, Theorem 2 and finally Theorem 1.

A preliminary note before we begin:

When we consider a (wlog) $n$ by $n$ channel $c$ of a convolutional layer, we assume it has shape

$$\begin{bmatrix} \boldsymbol{z}_{0,0} & \boldsymbol{z}_{1,2} & \cdots & \boldsymbol{z}_{0,n-1} \\ \boldsymbol{z}_{1,0} & \boldsymbol{z}_{2,2} & \cdots & \boldsymbol{z}_{1,n-1} \\ \vdots & \vdots & \ddots & \vdots \\ \boldsymbol{z}_{n-1,0} & \boldsymbol{z}_{n-1,1} & \cdots & \boldsymbol{z}_{n-1,n-1} \end{bmatrix}$$

| (a) | (b) | (c) | (d) |

Figure App.2: This figure visualizes the covariance matrix of one of the channels of a resnet trained on Imagenet. Black correspond to large values and white to small values. *(a)* we compute the covariance without a translation invariant dataset and without first preprocessing the images by DFT. We see that the covariance matrix is dense. *(b)* We compute the covariance after applying DFT, but without augmenting the dataset with translations. Even without enforcing translation invariance, we see that the covariance in the DFT basis is approximately diagonal. *(c)* Same as (a), but the dataset is augmented to be fully translation invariant. The covariance in the pixel basis is still dense. *(d)* Same as (c), but with dataset augmented to be translation invariant. The covariance matrix is exactly diagonal for a translation invariant dataset in a DFT basis.

When computing the covariance matrix however, we vectorize $c$ by stacking the columns under each other, and call the result $vec(c)$:

$$
vec(c) = \begin{bmatrix} \boldsymbol{z}_{0,0} \\ \boldsymbol{z}_{1,0} \\ \vdots \\ \boldsymbol{z}_{n-1,0} \\ \boldsymbol{z}_{0,1} \\ \vdots \\ \boldsymbol{z}_{n-1,n-1} \end{bmatrix} := \begin{bmatrix} \boldsymbol{z}_0 \\ \boldsymbol{z}_1 \\ \vdots \\ \boldsymbol{z}_{n-1} \\ \boldsymbol{z}_n \\ \vdots \\ \boldsymbol{z}_{n^2-1} \end{bmatrix}
$$

One useful identity when switching between these two notations (see e.g. [7]) is

$$
vec(AcB) = (B^T \otimes A)vec(c)
$$

where $A, B$ are matrices and $\otimes$ is the Kronecker product. A useful observation arising from this is:

**Lemma 3.** *The CCA vectors of $DFT(c_i), DFT(c_j)$ are the same (up to a rotation by $F$) as the CCA of $c_i, c_j$.*

*Proof:* From Section B we know that unitary transforms only rotate the $CCA$ directions. But while DFT pre and postmultiplies by $F, F^T$ – unitary matrices, we cannot directly apply this as the result is for unitary transforms on $vec(c_i)$. But, using the identity above, we see that $vec(DFT(c_i)) = vec(Fc_iF^T) = (F \otimes F)vec(c_i)$, which is unitary as $F$ is unitary. Applying the same identity to $c_j$, we can thus conclude that the DFT preserves CCA (up to rotations).

As Theorem 1 preprocesses the neurons with DFT, it is important to note that by the Lemma above, we do not change the CCA vectors (except by a rotation).

### C.1  Proof of Lemma 1

*Proof. Translation invariance is preserved* We show inductively that any translation invariant input to a convolutional channel results in a translation invariant output: Suppose the input to channel $c$, ($n$ by $n$) is translation invariant. It is sufficient to show that for inputs $X_i, X_j$ and $0 \leq a, b, \leq n-1$, $c(X_i) + (a, b) \mod n = c(X_j)$. But an $(a, b)$ shift in neuron coordinates in $c$ corresponds to a (height stride $\cdot a$, width stride $\cdot b$) shift in the input. And as $X$ is translation invariant, there is some $X_j = X_i + ($height stride $\cdot a$, width stride $\cdot b)$.

*cov(c) is circulant:*

Let $X$ be (by proof above) a translation invariant input to a channel $c$ in some convolution or pooling layer. The empirical covariance, $cov(c)$ is the $n^2$ by $n^2$ matrix computed by (assuming $c$ is centered)

$$\frac{1}{|X|} \sum_{X_i \in X} vec(c(X_i)) \cdot vec(c(X_i))^T$$

So, $cov(c)_{ij} = \frac{1}{|X|} \boldsymbol{z}_i^T \boldsymbol{z}_j = \frac{1}{|X|} \sum_{X_l \in X} \boldsymbol{z}_i^T(X_l) \boldsymbol{z}_j(X_l)$, i.e. the inner products of the neurons $i$ and $j$.

The indexes $i$ and $j$ refer to the neurons in their vectorized order in $vec(c)$. But in the matrix ordering of neurons in $c$, $i$ and $j$ correspond to some $(a_1, b_1)$ and $(a_2, b_2)$. If we applied a translation $(a, b)$, to both, we would get new neuron coordinates $(a_1 + a, b_1 + b), (a_2 + a, b_2 + b)$ (all coordinates mod $n$) which would correspond to $i + an + b \mod n^2$ and $j + an + b \mod n^2$, by our stacking of columns and reindexing.

Let $\tau_{a,b}$ be the translation in inputs corresponding to an $(a, b)$ translation in $c$, i.e. $\tau_{a,b} = $ (height stride$\cdot a$, width stride$\cdot b$). Then clearly $\boldsymbol{z}_{(a_1,b_1)}(X_i) = \boldsymbol{z}_{(a_1+a,b_1+b)}(\tau_{(a,b)}(X_i))$, and similarly for $\boldsymbol{z}_{(a_2,b_2)}$

It follows that $\frac{1}{|X|} \boldsymbol{z}_{(a_1,b_1)}^T \boldsymbol{z}_{(a_2,b_2)} = \frac{1}{|X|} \boldsymbol{z}_{(a_1+b,b_1+b)}^T \boldsymbol{z}_{(a_2+a,b_2+b)}$, or, with $vec(c)$ indexing

$$\frac{1}{|X|} \boldsymbol{z}_i^T \boldsymbol{z}_j = \frac{1}{|X|} \boldsymbol{z}_{(i+an+b \mod n^2)}^T \boldsymbol{z}_{(j+an+b \mod n^2)}$$

This gives us the circulant structure of $cov(c)$.

*$cov(c)$ is block circulant:* Let $\boldsymbol{z}^{(i)}$ be the $i$th column of $c$, and $\boldsymbol{z}^{(j)}$ the $j$th. In $vec(c)$, these correspond to $\boldsymbol{z}_{(i-1)n}, \ldots \boldsymbol{z}_{in-1}$ and $\boldsymbol{z}_{(j-1)n}, \ldots \boldsymbol{z}_{jn-1}$, and the $n$ by $n$ submatrix at those row and column indexes of $cov(vec(c))$ corresponds to the covariance of column $i, j$. But then we see that the covariance of columns $i+k, j+k$, corresponding to the covariance of neurons $\boldsymbol{z}_{(i-1)n+k\cdot n}, \ldots \boldsymbol{z}_{in-1+k\cdot n}$, and $\boldsymbol{z}_{(j-1)n+k\cdot n}, \ldots \boldsymbol{z}_{jn-1+k\cdot n}$, which corresponds to the 2-d shift $(1, 0)$, applied to every neuron. So by an identical argument to above, we see that for all $0 \le k \le n - 1$

$$cov(\boldsymbol{z}^{(i)}, \boldsymbol{z}^{(j)}) = cov(\boldsymbol{z}^{(i+k)}, \boldsymbol{z}^{(j+k)})$$

In particular, $cov(vec(c))$ is block circulant. $\square$

An example $cov(vec(c))$ with $c$ being 3 by 3 look like below:

$$\begin{bmatrix} A_0 & A_1 & A_2 \\ A_2 & A_0 & A_1 \\ A_1 & A_2 & A_0 \end{bmatrix}$$

where each $A_i$ is itself a circulant matrix.

## C.2   Proof of Lemma 2

*Proof.* This is a standard result, following from expressing a circulant matrix $A$ in terms of its diagonal form , i.e. $A = V \Sigma V^T$ with the columns of $V$ being its eigenvectors. Noting that $V = F$, the DFT matrix, and that vectors of powers of $\omega_k = \exp(\frac{2\pi i k}{n})$, $\omega_j = \exp(\frac{2\pi i k}{n})$ are orthogonal gives the result. $\square$

## C.3   Proof of Theorem 2

*Proof.* Starting with (a), we need to show that $cov(vec(DFT(c_i)), vec(DFT(c_i))$ is diagonal. But by the identity above, this becomes:

$$cov(vec(DFT(c_i)), vec(DFT(c_i)) = (F \otimes F)vec(c_i)vec(c_i)^T(F \otimes F)^*$$

By Lemma 1, we see that

$$cov(vec(c_i)) = vec(c_i)vec(c_i)^T = \begin{bmatrix} A_0 & A_1 & \dots & A_{n-1} \\ A_{n-1} & A_0 & \dots & A_{n-2} \\ \vdots & \vdots & \ddots & \vdots \\ A_1 & A_2 & \dots & A_0 \end{bmatrix}$$

with each $A_i$ circulant.

And so $cov(vec(DFT(c_i)), vec(DFT(c_i))$ becomes

$$\begin{bmatrix} f_{00}F & f_{01}F & \dots & f_{0,n-1}F \\ f_{10}F & f_{11}F & \dots & f_{1,n-1}F \\ \vdots & \vdots & \ddots & \vdots \\ f_{n-1,0}F & f_{n-1,1}F & \dots & f_{n-1,n-1}F \end{bmatrix} \begin{bmatrix} A_0 & A_1 & \dots & A_{n-1} \\ A_{n-1} & A_0 & \dots & A_{n-2} \\ \vdots & \vdots & \ddots & \vdots \\ A_1 & A_2 & \dots & A_0 \end{bmatrix} \begin{bmatrix} f_{00}^*F^* & f_{10}^*F^* & \dots & f_{n-1,0}^*F^* \\ f_{01}^*F^* & f_{11}^*F^* & \dots & f_{n-1,1}^*F^* \\ \vdots & \vdots & \ddots & \vdots \\ f_{0,n-1}^*F^* & f_{1,n-1}^*F^* & \dots & f_{n-1,n-1}^*F^* \end{bmatrix}$$

From this, we see that the $sj$th entry has the form

$$\sum_{l=0}^{n-1} \left( \sum_{k=0}^{n-1} f_{sk}FA_{l-k} \right) f_{lj}^*F^* = \sum_{k,l} f_{sk}f_{lj}^*FA_{l-k}F^*$$

Letting $[FA_rF^*]$ denote the coefficient of the term $FA_rF^*$, we see that (addition being $\mod n$)

$$[FA_rF^*] = \sum_{k=0}^{n-1} f_{sk}f_{(k+r)j}^* = \sum_k e^{\frac{2\pi isk}{n}} \cdot e^{\frac{-2\pi ij(k+r)}{n}} = e^{\frac{-2\pi ijr}{n}} \sum_{k=0}^{n-1} e^{\frac{2\pi ik(s-j)}{n}} = e^{\frac{-2\pi ijr}{n}} \cdot \delta_{sj}$$

with the last step following by the fact that the sum of powers of non trivial roots of unity are $0$.

In particular, we see that only the diagonal entries (of the $n$ by $n$ matrix of matrices) are non zero. The diagonal elements are linear combinations of terms of form $FA_rF^*$, and by Lemma 2 these are diagonal. So the covariance of the DFT is diagonal as desired.

Part (b) follows almost identically to part (a), but by first noting that exactly by the proof of Lemma 1, $cov(c_i, c_j)$ is also a circulant and block circulant matrix.

□

## C.4 Proof of Theorem 1

*Proof.* This Theorem now follows easily from the previous. Suppose we have a layer $l$, with channels $c_1, ..., c_k$. And let $vec(DFT(c_i))$ have directions $\tilde{\boldsymbol{z}}_0^{(i)}, \cdots \tilde{\boldsymbol{z}}_{n^2-1}^{(i)}$. By the previous theorem, we know that the covariance of all of these neurons only has non-zero terms $cov(\tilde{\boldsymbol{z}}_k^{(i)}, \tilde{\boldsymbol{z}}_k^{(j)})$.

So arranging the full covariance matrix to have row and column indexes being $\tilde{\boldsymbol{z}}_0^{(1)}, \tilde{\boldsymbol{z}}_0^{(1)}, \cdots \tilde{\boldsymbol{z}}_0^{(k)}, \tilde{\boldsymbol{z}}_1^{(1)} \cdots \tilde{\boldsymbol{z}}_{n^2}^{(k)}$ the nonzero terms all live in the $n^2$ $k$ by $k$ blocks down the diagonal of the matrix, proving the theorem. □

## C.5 Computational Gains

As the covariance matrix is block diagonal, our more efficient algorithm for computation is as follows: take the DFT of every channel ($n \log n$ due to FFT) and then compute covariances according to blocks: partition the $k^n$ directions into the $n^2$ $k$ by $k$ matrices that are non-zero, and compute the covariance, inverses and square roots along these.

A rough computational budget for the covariance is therefore $kn \log n + n^2 k^{2.5}$, while the naive computation would be of order $(kn^2)^{2.5}$, a polynomial difference. Furthermore, the DFT method also makes for easy parallelization as each of the $n^2$ blocks does not interact with any of the others.

Figure App.3: Learning dynamics per layer plots for conv (left pane) and res (right pane) nets trained on CIFAR-10. Each line plots the SVCCA similarity of each layer with its final representation, as a function of training step, for both the conv (left pane) and res (right pane) nets. Note the bottom up convergence of different layers

## D   Per Layer Learning Dynamics Plots from Section 4.1

## E   Additional Figure from Section 4.4

Figure App.4 compares the converged representations of two different initializations of the same convolutional network on CIFAR-10.

Figure App.4: Comparing the converged representations of two different initializations of the same convolutional architecture. The results support findings in [11], where initial and final layers are found to be similar, with middle layers differing in representation similarity.

## F   Experiment from Section 4.4

## G   Learning Dynamics for an LSTM

Figure App.5: Using SVCCA to perform model compression on the fully connected layers in a CIFAR-10 convnet. The two gray lines indicate the original train (top) and test (bottom) accuracy. The two sets of representations for SVCCA are obtained through 1) two different initialization and training of convnets on CIFAR-10 2) the layer activations and the activations of the logits. The latter provides better results, with the final five layers: pool1, fc1, bn3, fc2 and bn4 all being compressed to $0.35$ of their original size.

Figure App.6: Learning dynamics of the different layers of a stacked LSTM trained on the Penn Tree Bank language modeling task. We observe a similar pattern to that of convolutional architectures trained on image data: lower layer converge faster than upper layers.