[Reviews · NeurIPS 2017]

Reviewer 1



The paper presents an analysis of deep networks based on a combination of SVD and CCA to test similarity between representations at different layers. The authors use this technique to analyze aspects of the deep neural network training procedure, for example, how the representation builds bottom-up throughout training, and suggest an improved training procedure based on this analysis where lower-layers are frozen for most the training. The paper is well-written and contributions are clearly stated. The authors characterize the representation of a neuron by the vector of its activations for each data point, and then represent each layer as a matrix composed of the activation vectors for each neuron. Abstracting each layer as a kernel, and applying kernel SVD / CCA (with the linear kernel as a special case) would seem a more natural way to describe the same problem. In section 2.1, it is unclear why the authors perform CCA on two networks instead of PCA on a single network, except for the sake of expressing the analysis as a CCA problem. It would be good to have PCA as a baseline. While the bottom-up emergence of representation is an interesting effect, it is unclear whether the effect is general or is a consequence of the convnet structure of the model, in particular, the smaller number of parameters in the early layers. In Figure 4, it seems that the representations have relatively low correlation until the very end where they jump to 1. It suggests noise in the training procedure for the top layers. It would be interesting to test whether the frozen layers training strategy lead to higher correlations also for the higher layers.

Reviewer 2



The authors proposes a very nice and beautiful method. Their interpretation of a single neuron as a function is very influential. I strongly believe that with this interpretation one can understand deep networks and even further design much better ones by using tools from functional analysis. The paper is very well written and presented. Actually, the toy dataset represents the benefits of the experiments very clearly. The theoretical contributions are intuitive and makes their method applicable to CNNs. The experiments are convincing. Being all these said, I would like to see experiments on larger datasets.

Reviewer 3



This paper presents a novel technique for analyzing the learned representation of neural networks conditioned on the training or the validation data points. There are some shortcomings of the paper: 1) Although the proposed method is general, the authors only constrained their experiments to convolutional networks on a small task (CIFAR 10). They are encouraged to extend the experiments to fully connected networks, as well as going for larger and more interesting datasets (< 1M data points) using the proposed faster method. 2) How could the proposed method be of direct benefit for model building? For example: (1) Could the proposed method be used to suggest a freezing schedule during training? (2) Could SVCCA be used to determine how small a network could be to be able to approximate/compress a larger network? Minor comments: - From figure (2)-a: does lower layers need more directions to achieve better accuracies? - Check “Analysis of Deep Neural Networks with the Extended Data Jacobian Matrix” by Wang et al. from ICML 2016. They have a very similar objective and conclusions but a slightly different method. - In the model compression section, refer to prior art on using SVD weight compression, and papers on model compression/distillation. - The paper is too long, and some essential details are presented in the supplementary section. I suggest that the authors move the DFT method to the supplementary section to free up some space without impacting the paper’s readability.